# Genome-Wide Identification and Multi-Stress Response Analysis of the DABB-Type Protein-Encoding Genes in *Brassica napus*

**DOI:** 10.3390/ijms25115721

**Published:** 2024-05-24

**Authors:** Siyi Wang, Kunmei Wang, Qi Xia, Shitou Xia

**Affiliations:** Hunan Provincial Key Laboratory of Phytohormones and Growth Development, College of Bioscience and Biotechnology, Hunan Agricultural University, Changsha 410128, China; wangsiyi@stu.hunau.edu.cn (S.W.); yuhunan@stu.hunau.edu.cn (K.W.); xiaqi@stu.hunau.edu.cn (Q.X.)

**Keywords:** *B. napus*, *BnaDABBs*, synteny analysis, *S. sclerotiorum*, stress response

## Abstract

The DABB proteins, which are characterized by stress-responsive dimeric A/B barrel domains, have multiple functions in plant biology. In *Arabidopsis thaliana*, these proteins play a crucial role in defending against various pathogenic fungi. However, the specific roles of DABB proteins in *Brassica napus* remain elusive. In this study, 16 DABB encoding genes were identified, distributed across 10 chromosomes of the *B. napus* genome, which were classified into 5 branches based on phylogenetic analysis. Genes within the same branch exhibited similar structural domains, conserved motifs, and three-dimensional structures, indicative of the conservation of *BnaDABB* genes (*BnaDABBs*). Furthermore, the enrichment of numerous *cis*-acting elements in hormone induction and light response were revealed in the promoters of *BnaDABB*s. Expression pattern analysis demonstrated the involvement of *BnaDABBs*, not only in the organ development of *B. napus* but also in response to abiotic stresses and *Sclerotinia sclerotiorum* infection. Altogether, these findings imply the significant impacts of *BnaDABB*s on plant growth and development, as well as stress responses.

## 1. Introduction

*B. napus*, an important cultivated oilseed crop, not only provides edible vegetable oil resources for humans [1] but also serves as protein source for the animal feed industry [2]. However, the production of *B. napus* is frequently threatened by fungal diseases, among which Sclerotinia stem rot, caused by *S. sclerotiorum* [3], is particularly detrimental, impacting both yield and economic costs [4]. In the United States, economic losses exceeding $200 million annually have been attributed to *S. sclerotiorum* disease [5]. Similarly, annual economic losses linked to *S. sclerotiorum* disease have amount to 8.4 billion yuan in China [6]. Consequently, enhancing *B. napus* resistance to *S. sclerotiorum* is of paramount importance. In *A. thaliana*, *At*Dabb1 is predominantly located in the cytoplasm. Its expression is rapidly induced by salicylic acid (SA) and declines after 3 h, while jasmonic acid (JA) treatment leads to sustained high transcription levels for up to 24 h. Upon exposure to *Fusarium oxysporum*, the transcription of *AtDabb1* also increases and remains stable for between 6–12 h, possibly due to SA and JA accumulation post-fungal infection. Additionally, *At*Dabb1 exhibits significant antifungal effects, likely through interacting with fungal cell membrane components, and resulting in death of the fungal pathogen [7]. Another *A. thaliana* DABB protein, *At*HS1, possesses ribonuclease activity and notable antifungal properties. It effectively combats multiple pathogenic bacteria and fungi but does not hinder the growth of *Phytophthora infestans* and *Phytophthora nicotianae* lacking chitin in their cell walls, suggesting that *At*HS1 may disrupt microbial cell growth by interacting with cytoplasmic compounds [8]. Further studies indicate that the Olivetolic acid cyclase (OAC) in cannabis plays a crucial role in the biosynthesis of cannabinoids. This enzyme shares characteristics with DABB proteins and catalyzes the C2–C7 hydroxyaldehyde cyclization reaction of linear pentyl tetra-beta-ketide CoA as a substrate, leading to the production of olivetolic acid (OA) [9,10]. However, little is known currently about DABB proteins in *B. napus*.

Here, through whole-genome screening, 16 *BnaDABB*s were successfully identified, belonging to five putative groups (A–E). Detailed analyses were conducted on the conserved protein domains, motifs, and tertiary structure models of the *BnaDABBs*-encoded proteins, confirming the rational grouping of these genes, which are predominantly located on scaffold A01 and scaffold C01 chromosomes. Subcellular localization predictions indicated that these *Bna*DABBs function in multiple cellular regions, including chloroplasts, cytoplasm and nuclei. Additionally, the *cis*-elements, genomic collinearity, and physicochemical properties of these encoded proteins were investigated to further understand the roles of *Bna*DABBs in the growth, development, and stress responses of *B. napus*.

## 2. Results

### 2.1. Identification, Chromosomal and Subcellular Localization of DABBs in B. napus

When a BLASTP search was performed against the protein database of *B. napus* using *A. thaliana* DABB protein sequences (Q9SYD8/*At*1g51360 and Q9LUV2/*At*3g17210, antifungal proteins; Q9FK81/*At*5g22580, ABD19682.1 (*At*2g32500) and Q9SIP1/*At*2g31670, involved in stress response) as queries, 15 potential *BnaDABBs* were identified. Furthermore, using the hmmsearch tool, 17 potential *BnaDABBs* were identified. By removing the duplicate entries, 17 potential *BnaDABBs* were then identified. After verifying by the CD and SMART website, and removal of the *BnaDABB* gene with abnormal annotations, a total of 16 *BnaDABBs* were obtained subsequently (Appendix A, Text S1).

Members of the *BnaDABB* gene family were found to be distributed on chromosomes scaffold A01, scaffold A02, scaffold A03, scaffold A05, scaffold A08, scaffold A10, scaffold C01, scaffold C03, scaffold C05, and scaffold C09 respectively. The highest distribution was observed on chromosome C01, followed by chromosome A01. Based on their respective positions on the chromosomes, these *BnaDABBs* were named as *BnaA1DABB1, BnaA1DABB2, BnaA1DABB3*, *BnaA2DABB1*, *BnaA3DABB1*, *BnaA5DABB1*, *BnaA8DABB1*, *BnaA10DABB1*, *BnaC1DABB1, BnaC1DABB2, BnaC1DABB3, BnaC1DABB4*, *BnaC3DABB1, BnaC3DABB2*, *BnaC5DABB1*, and *BnaC9DABB1*, separately (Figure 1).

Subcellular localization prediction revealed that the majority of the *BnaDABBs* were localized in the chloroplast and cytoplasm. However, *BnaA1DABB1*, *Bna*A*3DABB1*, *BnaC1DABB1* and *BnaC3DABB1* were found to be located in the nucleus. Additionally, the subcellular localization of *BnaA10DABB1* and *BnaC9DABB1* was found to be widely distributed (Appendix A).

### 2.2. Analysis of Formation and Evolution of The BnaDABB Members in B. napus

To explore the evolution process of *BnaDABBs* in *B. napus*, we conducted a synteny analysis of the genomes of *B. napus*, *B. rapa*, and *B. oleracea* using the Python version of MCScan (JCVI toolkit 1.3.2). As shown in Appendix A, 13 pairs of *Br-BnaDABB* syntenic genes were found between *B. rapa* and *B. napus*, while there were 11 pairs of *Bo-BnaDABB* syntenic genes between *B. oleracea* and *B. napus*. *BnaDABBs* of the *Br-BnaDABB* syntenic gene pairs that displayed collinearity with the C genome of the *B. napus* were removed, resulting in the identification of 7 *BnaDABBs* originating from *B. rapa*. Similarly, from the *Bo-BnaDABB* syntenic gene pairs, 6 *BnaDABBs* originating from *B. oleracea* were obtained after excluding those that displayed collinearity with the A genome of the *B. napus* (Figure 2A). However, the number of these genes was lower than that of *BnaDABBs* identified from *B. napus*, suggesting the formation of new *BnaDABBs* in the later stages of evolution in *B. napus*.

Analysis of the duplications of *BnaDABBs* revealed that the *BnaA10DABB1* underwent segmental duplication, resulting in the formation of *BnaA02DABB1*. Similarly, the *BnaC1DABB2* underwent tandem duplication, leading to the generation of the *BnaC1DABB3* and *BnaC1DABB4* (Figure 2B). In a word, in the early evolution stages, *B. napus* inherited 13 *BnaDABBs* from its ancestors, *B. oleracea* and *B. rapa.* In the later stages, the formation of new *BnaDABBs* occurred through segmental and tandem duplication, ultimately resulting in the current 16 *BnaDABBs*.

In order to better understand the evolutionary overview of *BnaDABBs* in the *Brassicaceae* family, we further identified *DABBs* in *A. thaliana*, *B. carinata*, *B. juncea*, and *C. sativa* (Appendix A). Compared to *Arabidopsis*, the *Brassica* and *Camelina* genus exhibited a greater abundance of *DABBs*, suggesting an expansion of *DABBs* in *Brassica* and *Camelina* during their evolution (Figure 2C,D). Collinearity analysis of *BnaDABBs* in *Arabidopsis*, *Brassica*, and *Camelina* species revealed that the *BnaDABBs* exhibit widespread collinearity among them (Appendix A). Furthermore, it was shown that the majority of these genes are evolutionarily conserved (Figure 2E).

### 2.3. Phylogenetic Analysis and Biochemical Properties Calculation of BnaDABBs

Phylogenetic analysis indicated that the *Bna*DABBs in *B. napus* were diverged into five distinct branches during evolution. Based on this, we classified the *Bna*DABBs into five groups, designated as Group A, B, C, D, and E (Figure 3A). Conservative structural domain analysis revealed the presence of one DABB domain in both the A and B groups, whereas the C, D, and E groups contained two DABB domains (Figure 3B). The identification of conserved motifs in the *Bna*DABBs showed that all groups contained motif 1 and motif 2, with consistent types and quantities of motifs across the groups (Figure 3C, Appendix A). Further analysis revealed that the *Bna*DABBs sequences in Group E exhibited the smallest differences, followed by Group C. On the other hand, Groups A, B, and D showed greater differences in their *Bna*DABBs sequences (Appendix A). The 3D structure models of *Bna*DABBs further accentuated the differences among the groups. The *Bna*DABBs structures in Group B are the simplest, characterized by a small number of α helices, β sheets, β turns, and random coils. In contrast, the 3D structures in Groups A, C, D, and E are more complex, consisting of multiple α helices, β sheets, β turns and random coils (Figure 3D). Additionally, similar differences were observed in the DABB domains (Figure 3E). However, within each group, the *Bna*DABB proteins exhibited high similarity in 3D structures and domains, even in the presence of sequence variations (Figure 3D,E).

Furthermore, the protein with the largest molecular weight was found in group E, while group A exhibited the smallest molecular weight; the theoretical isoelectric point (pI) ranged from 5.12 to 8.76. Apart from group C, all groups were stable proteins. The aliphatic index of group E proteins fell in between 77 and 80, which was below 100 and lower than the aliphatic index values of other groups. Additionally, the grand average of hydrophilicity was −0.286, approaching zero. These results suggest that the group E proteins have better hydrophobicity compared to other groups (Table 1), and the *Bna*DABB proteins may possess diverse functions in *B. napus*.

### 2.4. Analysis of Cis-Elements and Expression Patterns of BnaDABBs

*Cis*-acting regulatory elements act as molecular switches in the transcriptional regulation of a dynamic network of gene activities that govern diverse biological processes, encompassing abiotic stress responses, hormone signaling, and developmental processes. Therefore, we analyzed the *cis*-elements of *BnaDABBs* promoters by PlantCARE (Appendix A). The enriched *cis*-regulatory elements in the 2000 bp promoter region suggests that *BnaDABBs* may be involved in multiple biological processes such as light response (Figure 4A), hormone response (Figure 4B), binding site (Figure 4C), low temperature (Figure 4D), and so on (Figure 4E). Of particular interest is the potential role of *BnaDABBs* in hormone response. We then analyzed the functionality of the promoter regions of *BnaDABBs* in relation to hormone-response-associated *cis*-elements. The results indicated that a majority of *BnaDABBs* contained the CGTCA-motif and TGACG-motif, suggesting their potential association with MeJA-responsiveness. Additionally, we found that *BnaDABBs* comprised various hormone-response-related *cis*-elements, indicative of their significance in hormone responses (Figure 4B).

To further investigate the putative functions of *BnaDABBs* in plant growth and development, we analyzed the tissue expression patterns of *BnaDABBs* (Appendix A), and found that *BnaA1DABB1* and *BnaC1DABB1* from Group C exhibited higher expression levels in various stages of leaf and seed development, as well as in roots and stems. *BnaA1DABB3* and *BnaC1DABB4* from Group A exhibited similar expression patterns, but their expression levels were lower compared with *BnaA1DABB1* and *BnaC1DABB1*. Additionally, *BnaA10DABB1* and *BnaC9DABB1* from Group B showed higher expression levels during leaf development from 1–11 days, as well as in stems (Appendix A). These findings further underscore the potential multifunctionality of *BnaDABBs* in *B. napus*.

### 2.5. Expression Profiling of BnaDABBs under Plant Hormones Treatment

To reveal the relationship between *BnaDABBs* in *B. napus* and plant hormones, the expression patterns upon various plant hormones treatment were investigated (Appendix A). Expression profiling of *BnaDABB* genes in leaf and root indicated that the majority of these genes showed no significant differences in expression levels compared to the control group after treated with phytohormones IAA, GA, ABA, and JA. However, in leaf, *BnaC1DABB2/3/4* in group A, *BnaA10/C9DABB1* in group B, and *BnaA1/C1DABB1* in group C exhibited increased expression levels after JA treatment (Appendix A). In roots, *BnaC1DABB2/3* in group A, *BnaC9DABB1* in group B, and *BnaA1/C1DABB1* in group C also displayed similar expression patterns (Appendix A). Furthermore, *BnaA10DABB1* in group B showed decreased expression levels after ABA treatment in leaf (Appendix A), and *BnaA8DABB1* in Group D exhibited a similar expression pattern in root (Appendix A). Overall, these expression results suggest that *BnaDABBs* may play a role in response to JA and ABA.

### 2.6. Expression Profiling of BnaDABB Genes under Abiotic Stresses Treatment

To elucidate the expression patterns of *BnaDABBs* in abiotic stress responses in *B. napus*, we investigated the expression of these genes in leaf and root after salt, drought, freezing, low temperature, high temperature, and osmotic stress treatments (Appendix A). It was revealed that compared to the control group of *BnaDABBs* in leaf, *BnaA1DABB3* and *BnaC1DABB4* in group A exhibited downregulated expression levels after salt, drought, low temperature, high temperature and osmotic stress treatments; *BnaA10DABB1* and *BnaC9DABB1* in group B showed downregulated expression levels after salt, freezing, and osmotic stress treatments (Appendix A). Compared with the control group of *BnaDABBs* in root, *BnaC1DABB2/3* in group A displayed downregulated expression levels after salt, drought, freezing, high temperature, and osmotic stress treatments. *BnaA10DABB1* and *BnaC9DABB1* in group B exhibited downregulated expression levels after salt, freezing, low temperature, high temperature, and osmotic stress treatments, but upregulated expression levels after drought treatment (Appendix A). These results suggest that *BnaDABBs* also play an important role in the biological processes of non-biological stress responses in *B. napus*.

### 2.7. Expression Profiling of BnaDABBs during Infection by Sclerotinia sclerotiorum

To uncover the importance of *BnaDABBs* during the interaction between *B. napus* and *S. sclerotiorum*, we conducted an analysis of the expression levels of *BnaDABBs* in leaves of susceptible (*B. napus* cv. Westar, Westar) and middle tolerant (*B. napus* cv. ZhongYou 821, ZY821) varieties after 24 h of pathogen exposure (Figure 5A, Appendix A). The examination unveiled an upregulation of 2 *BnaDABBs* and a downregulation of 7 *BnaDABBs* in ZY821, while in Westar, 7*BnaDABBs* were downregulated. Moreover, compared to Westar, *BnaCnng51720D* and *BnaA01g23550D* were upregulated, whereas *BnaAnng40030D* was downregulated in ZY 821. Additionally, we also observed a downregulation in the expression levels of six shared *BnaDABBs* in both ZY 821 and Westar (Figure 5B). These findings suggest that these genes, *BnaCnng51720D*, *BnaA01g23550D* and *BnaAnng40030D*, may contribute to the resistance of *B. napus* against *S. sclerotiorum*.

As tolerant gene expression in varieties might be similar, and there remains a notable scarcity of data on the interaction between ZS11, a commonly used *B. napus* variety and *S. sclerotiorum* [11,12,13], we thus verified the highly homologous *BnaDABBs* in ZS11 (Appendix A) to gain a deeper understanding of their expression characteristics of *BnaCnng51720D* (*BnaC1DABB1*), *BnaA01g23550D* (*BnaA1DABB1*), and *BnaAnng40030D* (*BnaA1DABB2*) after 24 and 48 h of *S. sclerotiorum* infection using qPCR. The results indicated that at 24 h post-infection, there were no significant changes in the expression levels of *BnaC1DABB1* and *BnaA1DABB1*, while the expression of *BnaA1DABB2* significantly decreased (*p* < 0.01), consistent with the expression pattern observed in ZY 821. However, after 48 h of infection, the expression levels of these genes significantly increased, showing a notable change compared to 24 h (Figure 5C–E). These findings suggest that *BnaC1DABB1*, *BnaA1DABB1*, and *BnaA1DABB2* may function differently in ZY82 and ZS11. The divergence may be associated with the distinct promoter sequences of these genes. To further validate this hypothesis, we conducted a comparative analysis of the promoter sequences of *BnaC1DABB1*, *BnaA1DABB1*, and *BnaA1DABB2* in ZS11 and *B. napus*, and detected some differences in the 2000bp promoter sequences of these genes (Appendix A). Compared to ZY821, *BnaA1DABB1* of ZS11 exhibited lower conservation and significant differences within the promoter sequence spanning 1 to 314 bp (Figure 6A). Similarly, *BnaA1DABB2* of ZS11 showed reduced conservation and marked sequence divergence in the promoter region spanning 1 to 650 bp compared with ZY821 (Figure 6B). In addition, *BnaC1DABB2* of ZS11 displayed lower conservation and substantial sequence differences in the promoter regions spanning 1 to 95 bp, 693 to 724 bp, and 1800 to 1895 bp compared to ZY821(Figure 6C).

## 3. Discussion

It is believed that *B. napus* (AACC, 2n = 38) originated from natural hybridization between *B. rapa* (AA, 2n = 20) and *B. oleracea* (CC, 2n = 18) more than 7,500 years ago [14]. Building upon this premise, we investigated the evolutionary processes underlying the formation of the *BnaDABB* gene family. Our analyses revealed that during evolution, *B. napus* retained *7 BnaDABBs* from *B. rapa* and 6 *BnaDABBs* from *B. oleracea*. Furthermore, we found that tandem and segmental duplication are significant mechanisms driving the expansion of the *BnaDABB* gene family. In addition to the 13 *BnaDABBs* inherited from *B. rapa* and *B. oleracea*, *B. napus* acquired 3 additional *BnaDABBs* through these duplication events, resulting in the current set of 16 *BnaDABBs*. These results provide a deep understanding of the evolutionary history of the *BnaDABB* gene family.

*A. thaliana* [15], *B. carinata* [16], *B. juncea* [17], and *C. sativa* [18] are important species within the Brassicaceae family. Through synteny analysis to explore the evolutionary relationships of *BnaDABB* gene families in these species, we observed a widespread synteny relationship between *B. napus* and *A. thaliana*, *B. carinata*, *B. juncea*, and *C. sativa*. Compared to Arabidopsis, the *DABB* gene members in *B. carinata, B. juncea*, *C. sativa*, and *B. napus* exhibited an expanding trend during evolution. Moreover, we noticed that the majority of *BnaDABBs* showed conservation in evolution. These findings collectively suggest that *DABB* gene members may play significant biological roles in *Brassicaceae* species.

Plant hormones are critical signaling molecules that play essential roles in regulating plant growth, development, and responses to environmental stress [19,20]. Analysis of the *cis*-regulatory elements in the promoter regions of *BnaDABBs* revealed that the majority of *BnaDABBs* contain at least two or more hormone-responsive *cis*-elements, suggesting that *BnaDABBs* may function in hormone responses. To confirm this hypothesis, we analyzed the expression profiles of *BnaDABBs* under multiple hormone stresses. As expected, we found that in *B. napus* leaves and roots under jasmonic acid (JA) stress, the expression levels of *BnaC1DABB2/3*, *BnaC9DABB1*, and *BnaA1/C1DABB1* significantly increased compared with the control group. Additionally, we observed that *BnaA10DABB1* and *BnaA8DABB1* might be associated with abscisic acid (ABA) stress. Under ABA stress, the expression of *BnaA10DABB1* and *BnaA8DABB1* significantly decreased. These findings suggest a potential yet unknown functional link between *BnaDABBs* and JA or ABA.

Research has confirmed that *DABB* genes may play a crucial role in abiotic stress responses. For instance, the *SP1* [21] gene from Aspen shows responsiveness to various stress conditions such as salt, cold, and heat. Similarly, we found that under salt, drought, freezing, low temperature, high temperature, and osmotic stresses, the expression levels of *BnaA1DABB3*, *BnaC1DABB4*, *BnaA10DABB1*, and *BnaC9DABB1* were downregulated compared to the control group. Additionally, the expression levels of *BnaA1DABB3* and *BnaC1DABB4* in leaves were downregulated after drought, low-temperature, and high-temperature treatments. When treating plant roots in the same manner, the expression levels of *BnaC1DABB2/3* were downregulated after salt, drought, freezing, high-temperature, and osmotic stress treatments. Moreover, the expression levels of *BnaA10DABB1* and *BnaC9DABB1* were downregulated after salt, freezing, low-temperature, and high-temperature treatments but upregulated after drought treatment.

It was further observed that, in addition to plant hormones and abiotic stresses, *BnaDABBs* also responded to the infection of *S. sclerotiorum*. Within 24 h post-infection (hpi), the expression levels of most *BnaDABBs* were downregulated, yet by 48 hpi, the expression levels of *BnaA1DABB2*, *BnaA10DABB1*, and *BnaA3DABB1* were significantly upregulated compared to 24 hpi, indicating a potential role of *BnaDABBs* in the interaction between *S. sclerotiorum* and rapeseed. This interaction might be regulated by methyl MeJA, as most *BnaDABBs* contain *cis*-acting elements responsive to MeJA, such as TGACG-motif and CGTCA-motif. Previous studies have shown that methyl MeJA can induce the expression of disease resistance genes in plants and play a critical role in plant disease resistance [22]. Additionally, research on *DABB* genes in *Arabidopsis* suggests that the expressed proteins exhibit antifungal activity and belong to a novel class of antifungal proteins [7,8]. Hence, it is plausible to postulated that *DABB* genes in rapeseed, induced by MeJA, may encode proteins with antifungal activity, thereby enhancing resistance against pathogenic fungi. However, further in-depth studies are required to validate the antifungal proteins encoded by *DABBs* in rapeseed.

In summary, our research indicates that the *DABB* genes have diverged into five evolutionarily conserved branches in rapeseed, which have roles in various hormone responses, abiotic stress, and biotic stress responses, providing a new understanding of *DABBs* in plants for further exploration of the functions of *BnaDABBs* in *B. napus*.

## 4. Materials and Methods

### 4.1. Fungal Strains and Culture Conditions

The wild-type strain *S. sclerotiorum* 1980 was grown on potato dextrose agar plates (potato dipping powder 5 g/L, glucose 20 g/L, agar 15 g/L, and chloramphenicol 0.1 g/L, Bio-Way Technology, Shanghai, China), and cultured in a constant temperature incubator at 20 °C.

### 4.2. Plant Materials and Growth Conditions

Seeds of *B. napus* (Zhongshuang 11, ZS11) were sterilized and put in a refrigerator at 4 °C for vernalization for 2–3 days, then planted on soil and cultivated in a growth room at a temperature of 22 °C, with a photoperiod of 16 h of light followed by 8 h of darkness. Following a 5-week cultivation period, the method described by previous researchers [12] was utilized to conduct the *S. sclerotiorum* infection.

### 4.3. Identification of DABB Genes in B. napus and Related Species

A BLASTP (2.12.0) [23] was conducted using the *At*DABB protein sequence from *A. thaliana* against the *B. napus* (ZS11.v0) protein database, employing an e-value threshold of 1 × 10^−10^. Moreover, the Dabb domain hmm file (PF07876.hmm) was downloaded from the Pfam website [24] (http://pfam.xfam.org/, accessed on 11 October 2023), and the hmmsearch tool (HMMER 3.3.2) [25] was employed to identify potential genes harboring this conserved domain in the *B. napus* protein database. Duplicate genes were eliminated by comparing the outcomes of the two distinct searches. Subsequently, the remaining genes were subjected to further validation for the presence of the DABB-type domain using the CD search website [26] (https://www.ncbi.nlm.nih.gov/Structure/cdd/wrpsb.cgi, accessed on 11 October 2023) and the SMART website [27] (http://smart.embl-heidelberg.de/,accessed on 11 October 2023). By identifying the *Bna*DABB protein sequence, a parallel approach was employed to search for DABB proteins in *B. carinata*, *B. juncea*, *B. rapa*, *B. oleracea*, and *C. sativa*.

The *At*DABB family protein sequences information can be found in Appendix A, and the protein and genome database of *B.napu*s.ZS11.v0 *and B.carinata.*zd-1.v0, were downloaded from the BnIR website (https://yanglab.hzau.edu.cn/BnIR, accessed on 10 October 2023) [28]. *The A. thaliana. TAIR10.57*, *B.juncea*.ASM1870372v1.57, *C.sativa*.Cs.57, *B.oleracea*.BOL.57 and *B.rapa*.Brapa_1.0.57 protein and genome database was downloaded from the EnsembI Plants website (http://plants.ensembl.org/index.html, accessed on 10 October 2023).

### 4.4. Chromosomal Location, Subcellular Localization and Collinearity Analysis of BnaDABB Genes

To ascertain the chromosomal locations of the *BnaDABBs* in *B. napus*, the generic feature format (gff3) files and formatted sequence (fa) files of the *B. napus* genome were utilized to acquire positional data and the length of the chromosomes housing the *BnaDABB* genes. Subsequently, chromosome mapping was conducted employing the visualization tool MG2C (v2.1) [29]. The genome database, CDS database, and gff3 file were downloaded from the BnIR website (https://yanglab.hzau.edu.cn/BnIR/genome_data, accessed on 10 October 2023). The *BnaDABBs* were designated based on their respective positions on the chromosomes. Furthermore, the subcellular localization of the identified *Bna*DABB family members was assessed using Plant-mPLoc [30] (http://www.csbio.sjtu.edu.cn/bioinf/plant-multi/, accessed on 11 October 2023). The Python version3.9.16 of MCScan (JCVI toolkit 1.3.2) [31] was utilized to investigate the intra- and inter-species collinearity relationships of the *BnaDABB* gene family.

### 4.5. Phylogenetic Analysis and Biochemical Properties Calculation

Multiple sequences alignment of the *Bna*DABB proteins and promoter sequences were performed using the Clustal Omega program [32] (https://www.ebi.ac.uk/Tools/msa/clustalo/, accessed on 16 October 2023). Then, a phylogenetic tree was constructed via the neighbor-joining (NJ) method using MEGA 11 [33] software with 1000 bootstrap repetitions. The multiple sequences alignment results were generated by the visualization tool Jalview 2.11.3.2. The software MEME5.5.1 [34] was used to investigate the conserved motifs of *Bna*DABB proteins; the parameters were set with the maximum number of motifs as 11 and the motif width as 6–100 amino acids. The phylogenetic tree and conserved domain were visualized using iTOL V6 software [35] (https://itol.embl.de, accessed on 16 October 2023). The *BnaDABB* gene family members were subjected to 3D model construction using the homology modeling software SWISS-MODEL (https://swissmodel.expasy.org, accessed on 16 October 2023) [36]. The constructed models were subsequently evaluated using the online software SAVES (https://saves.mbi.ucla.edu, accessed on 16 October 2023). The 3D structure of the protein domain of BnaDABBs was visualized using VMD1.9.4 software [37], and the *Bna*DABB protein biochemical properties were predicted using the ExPASy ProtParam tool [38] (https://web.expasy.org/protparam/, accessed on 16 October 2023).

### 4.6. Analysis of Cis-Element, Expression Patterns and Abiotic Stress

To identify the *cis*-element of *BnaDABBs*, TBtools2.030 [39] was used to obtain the 2000 bp sequences of the genomic promoter. Then, the PlantCARE [40] was used to predict the *cis*-elements on these promoters. Thus, the number and types of different *cis*-acting elements in *BnaDABBs* were classified and visualized with excel. The expression patterns of *BnaDABBs* in different tissues and other abiotic stress treatment were obtained from the BnTIR (https://yanglab.hzau.edu.cn/BnTIR, accessed on 20 October 2023) database [28]. The heatmaps were drawn by TBtools2.030 [39].

### 4.7. RNA Extraction and RT–qPCR Analysis

Fungal mycelial plugs with a diameter of 5 mm were obtained from the *S. sclerotiorum*. These plugs were inoculated onto the leaves of rapeseed plants and covered with a layer of cling film to maintain the moisture of the fungal plugs. Leaf tissues within 1 cm of the infection site were collected immediately into liquid nitrogen after 24 h and 48h post-infection. This experiment was replicated three times. For the expression analysis of *BnaDABBs*, total RNA of the 0, 24, 48 post inoculation was isolated from the infected leaves using the Eastep™ Super Total RNA Extraction Kit (Promega, Madison, WI, USA). Reverse transcription was carried out using the GoScript™ Reverse Transcription System (Promega, Beijing, China). The RT-qPCR assay was carried out using 2 × SYBR Green Premix Pro Taq HS Premix (AG11702, Accurate Biotechnology (Hunan) Co., Ltd., Changsha, China) and a Step-One real-time fluorescence PCR instrument (Applied Biosystems, Foster City, CA, USA). The RT-qPCR reaction system contained 10 ng cDNA, 4 µM of each primer, 5 µL 2 × SYBR Green Premix Pro Taq HS Premix, 0.2 µL ROX reference dye, and 3.4 µL RNase-free water. The RT-qPCR programming was as follows: denaturation at 95 °C for 2 min, followed by 40 cycles (95 °C for 20 s, 55 °C for 20 s, and 72 °C for 30 s). *BnaActin7* (*BnaC09T0560200ZS*) was used as the reference gene. Two or more independent biological replicates and three technical replicates of each sample were performed for quantitative PCR analysis, and the 2^−ΔΔCt^ algorithm was used to analyze the results [39].

The RNA-seq data (Accession Number: GSE81545) was downloaded from NCBI GEO database (https://www.ncbi.nlm.nih.gov/geo/query/acc.cgi?acc=GSE81545, accessed on 20 November 2023). Gene-specific primers used in the experiments are listed in Appendix A.

## Figures and Tables

**Figure 1 ijms-25-05721-f001:**
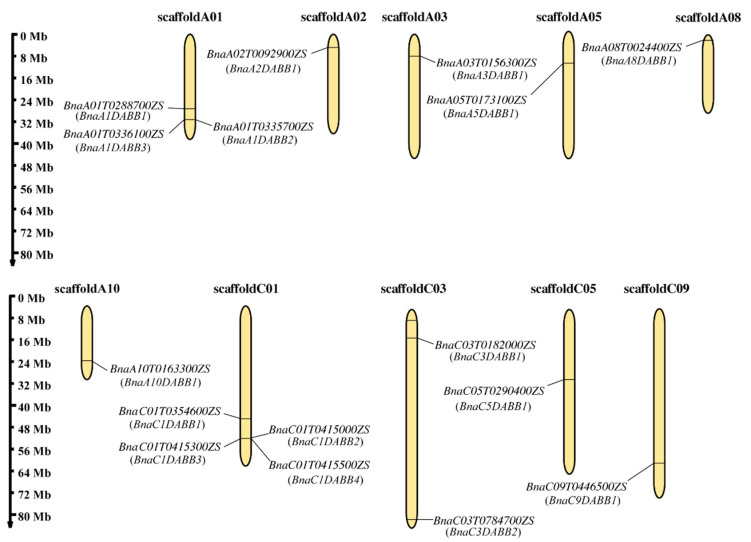
Distributions of *BnaDABBs* on the chromosomes of *B. napus*. Scaffold A or scaffold C represents A or C subgenome of *B. napus* respectively. The chromosome numbers are shown at the tops of the Scaffold. *BnaDABBs* names and IDs were labeled at the left or right of the chromosomes. Scale bars on the left indicate the chromosome lengths (Mb).

**Figure 2 ijms-25-05721-f002:**
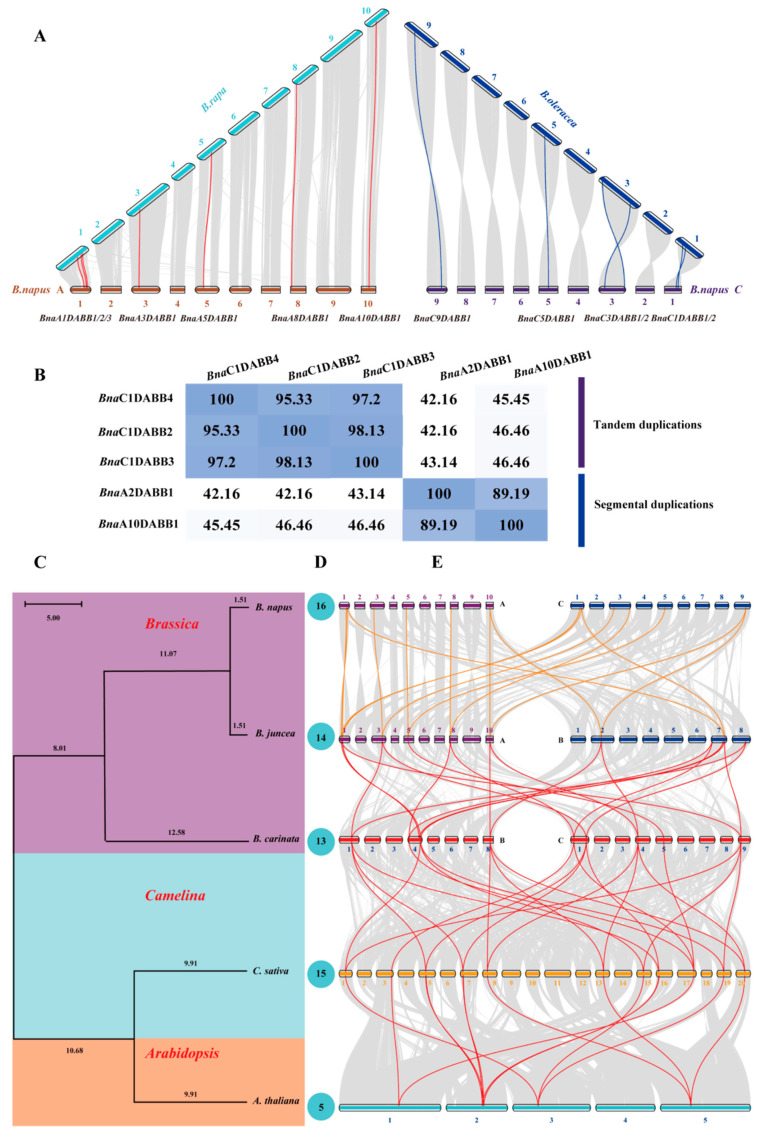
Evolutionary analysis of the BnaDABBs: (**A**) collinearity analysis of the genomes of B. napus, B. rapa, and B. oleracea. Grey lines represent syntenic sequences, and the highlighted red and blue lines indicated syntenic gene pairs of BnaDABBs; (**B**) analysis of segmental and tandem duplication of the BnaDABBs. The values indicate identity (%); (**C**) phylogenetic tree analysis of B. napus, A. thaliana, B. carinata, B. juncea and C. sativa by time tree (http://www.timetree.org, accessed on 11 October 2023); (**D**) the number of DABBs presented in B. napus, A. thaliana, B. carinata, B. juncea and C. sativa; and (**E**) collinearity analysis of the genomes of B. napus, A. thaliana, B. carinata, B. juncea, and C. sativa. The chromosome number was labeled at the top or bottom of each chromosome. The colored bars represent the chromosomes of the different species. The grey lines in the background indicate the collinear blocks in the genomes of the two species connected by the grey lines, while the colored lines highlight the syntenic DABB gene pairs.

**Figure 3 ijms-25-05721-f003:**
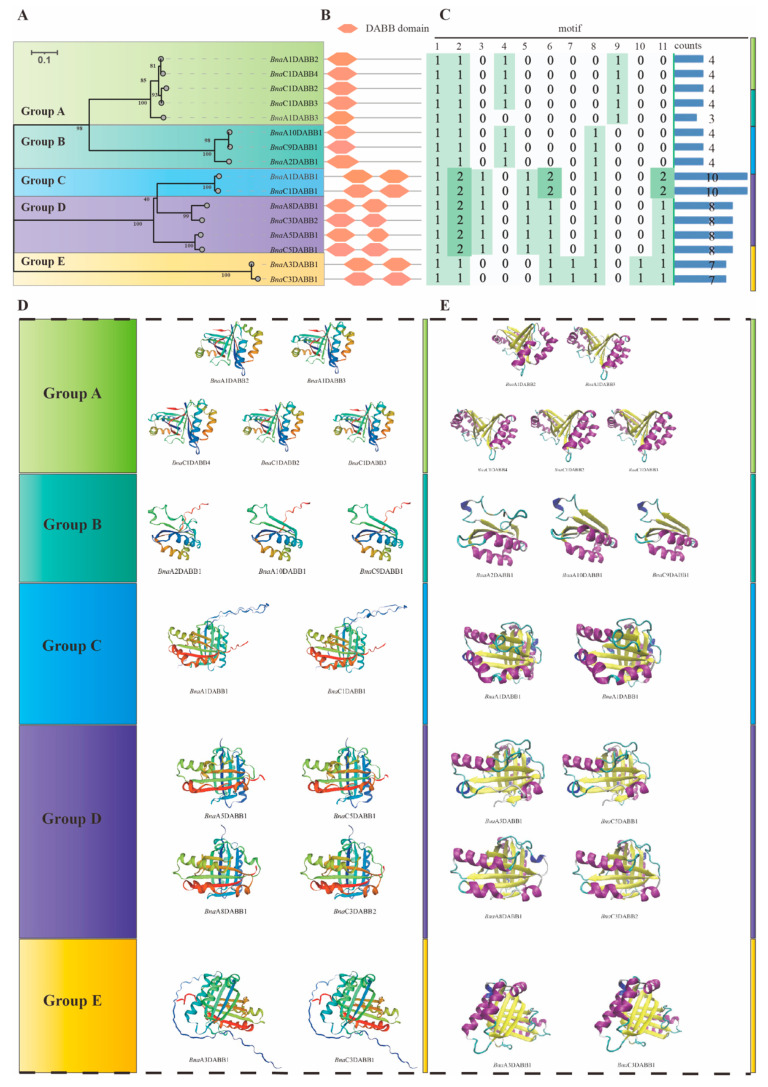
Phylogenetic, protein sequences and structure analysis of the *Bna*DABB proteins: (**A**) phylogenetic analysis; (**B**) conservative structural domain analysis; (**C**) conservative motif identification. The quantity of various types of motifs was represented by a green color scale, with a color range from minimum to maximum. The larger the value, the darker the color; and (**D**) A comparative evaluation was carried out on the three-dimensional protein structures of *Bna*DABBs in ZS11. The SWISS-MODEL was employed in the construction of the 3D protein structure models of *Bna*DABBs, and visualized using the Rainbow Model color scheme. (**E**) A contrastion was made on the 3D models of DABB structural domains of *Bna*DABB proteins in ZS11. The distinct secondary protein structures were represented using different colors in the visualization process of the DABB-type domains of *Bna*DABB proteins, which was executed through VMD.

**Figure 4 ijms-25-05721-f004:**
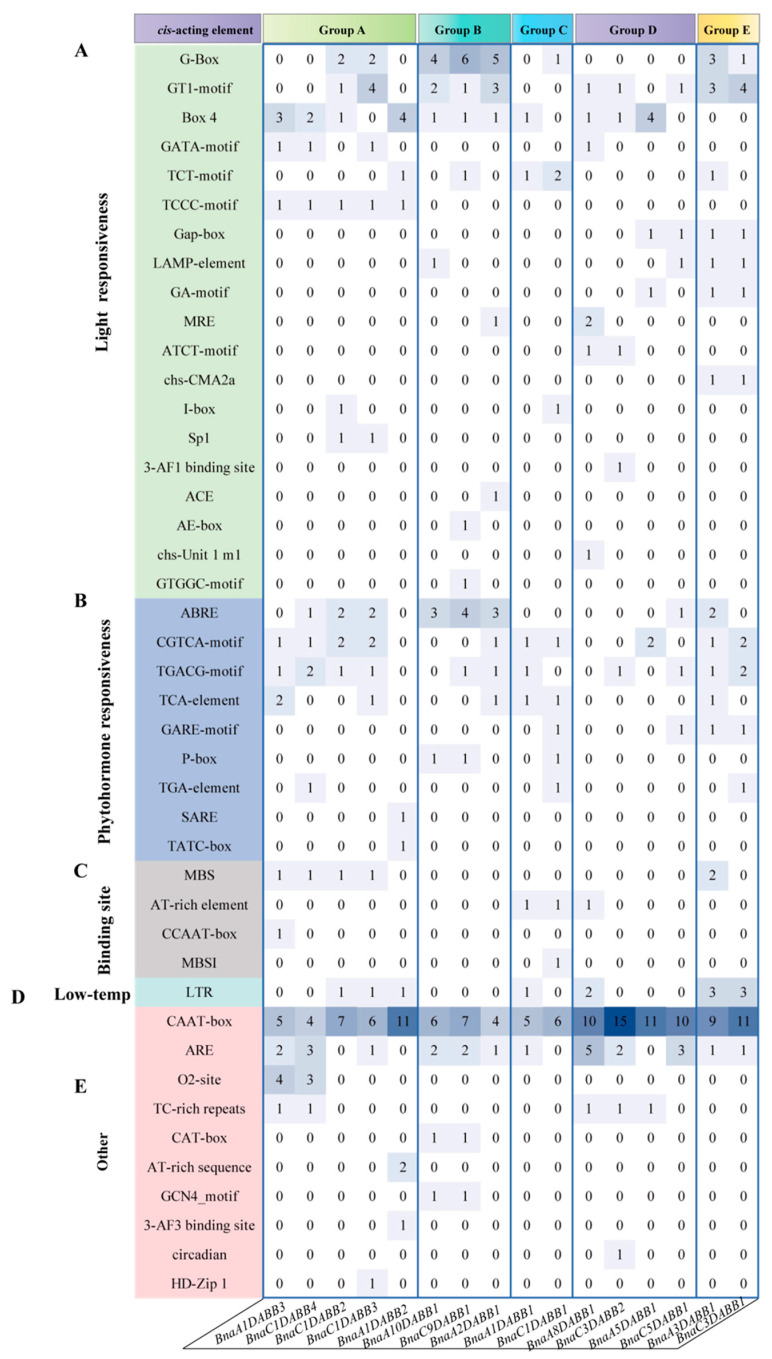
Analysis of *cis*-element and expression patterns: (**A**) *Cis*-elements associated with light responsiveness; (**B**) *Cis*-elements associated with phytohormone responsiveness; (**C**) *Cis*-elements associated with binding site; (**D**) *Cis*-elements associated with light responsiveness; and (**E**) other *cis*-elements.

**Figure 5 ijms-25-05721-f005:**
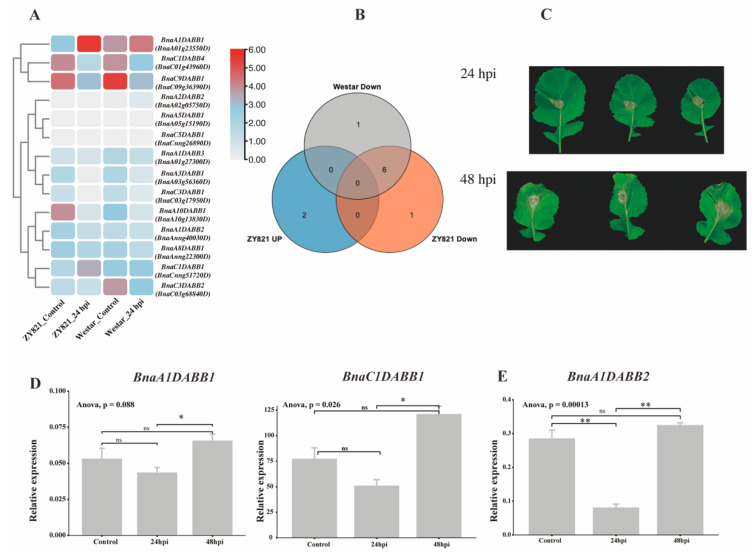
Analysis of expression profiling of *BnaDABBs* during *S. sclerotiorum* infection: (**A**) expression profiling of *BnaDABBs* during *S. sclerotiorum* infection in *B. napus* of ZY821 and Westar at 24 hpi; (**B**) number of shared and *BnaDABB* genes between ZY821 and Westar; (**C**) lesions induced by *S. sclerotiorum* on the leaves of ZS11 following inoculation in *B. napus*; (**D**) expression profiling of *BnaC1DABB1*, *BnaA1DABB1* during *S. sclerotiorum* infection in ZS11 at 24 and 48 hpi; and (**E**) expression profiling of *BnaA1DABB2*during *S. sclerotiorum* infection in ZS11 at 24 and 48 hpi.The experiment was conducted three times with similar results. Error bars represent the standard deviation (SD). The expression data was subjected to statistical analysis and visualization using the R package ggpubr (https://rpkgs.datanovia.com/ggpubr/, accessed on 10 January 2024). One-way analysis of variance (ANOVA) and Student’s *t*-test were employed to determine the significant differences among the control group, 24 h post-infection (24 hpi), and 48 h post-infection (48 hpi) Student’s *t*-test (ns indicates no significant difference; * indicates significant difference at 0.05 level, *p* < 0.05; ** indicates significant difference at 0.01 level, *p* < 0.01).

**Figure 6 ijms-25-05721-f006:**
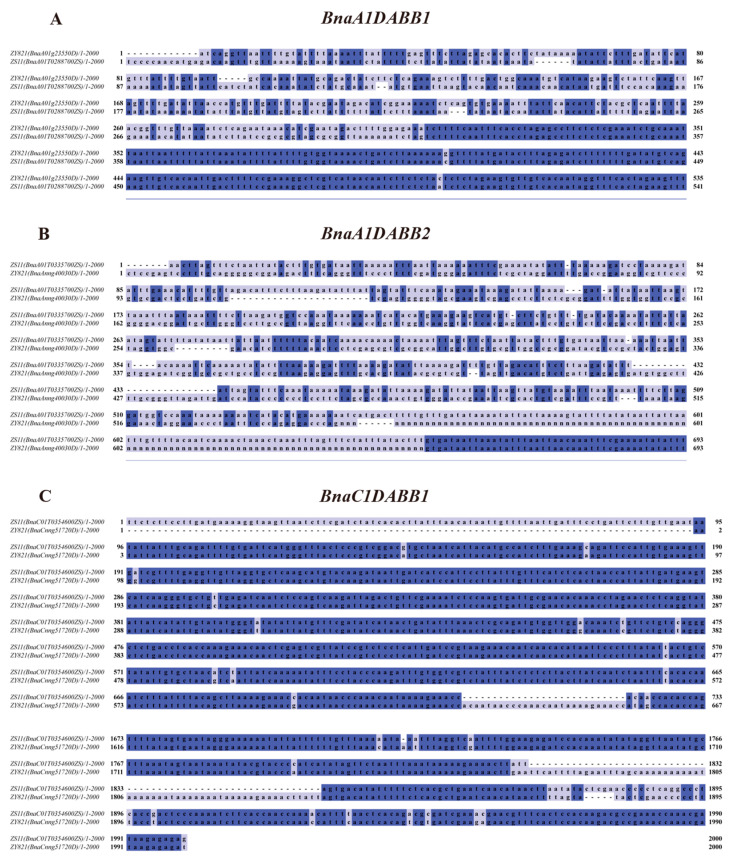
Promoter sequence analysis: (**A**) Promoter sequence comparison of the *BnaA1DABB1* gene between species ZS11 and ZY821; (**B**) Promoter sequence comparison of the *BnaA1DABB2* gene between species ZS11 and ZY821; and (**C**) Promoter sequence comparison of the *BnaC1DABB1* gene between species ZS11 and ZY821. Color represents the degree of conservatism of sequence loci, with darker colors indicating greater conservatism.

**Table 1 ijms-25-05721-t001:** Physio-biochemical characterization of *Bna*DABB family members.

Group	Name	Gene ID	Molecular Weight	Theoretical pI	Instability Index	Aliphatic Index	Grand Average of Hydropathicity
A	*Bna*A1DABB2	*BnaA01T0335700ZS*	11,213.85	5.46	19	93.43	−0.297
A	*Bna*A1DABB3	*BnaA01T0336100ZS*	12,142.96	5.41	26.72	95.61	−0.103
A	*Bna*C1DABB2	*BnaC01T0415000ZS*	12,142.96	5.42	24.86	94.67	−0.123
A	*Bna*C1DABB3	*BnaC01T0415300ZS*	12,147.98	5.26	25.83	95.61	−0.106
A	*Bna*C1DABB4	*BnaC01T0415500ZS*	12,115.94	5.41	27.42	95.61	−0.077
B	*Bna*A2DABB1	*BnaA02T0092900ZS*	13,701.93	8.76	35.92	82.07	−0.146
B	*Bna*A10DABB1	*BnaA10T0163300ZS*	12,321.08	5.22	22.95	85.95	0.109
B	*Bna*C9DABB1	*BnaC09T0446500ZS*	12,351.11	5.22	23.72	85.95	0.105
C	*Bna*A1DABB1	*BnaA01T0288700ZS*	29,132.53	7.14	45.9	98.52	0.051
C	*Bna*C1DABB1	*BnaC01T0354600ZS*	29,343.92	7.14	48.98	99.93	0.054
D	*Bna*A5DABB1	*BnaA05T0173100ZS*	22,568.01	4.85	33.16	107.33	0.073
D	*Bna*A8DABB1	*BnaA08T0024400ZS*	22,540.81	5.17	30.86	101.67	−0.085
D	*Bna*C3DABB2	*BnaC03T0784700ZS*	22,363.70	5.49	28.18	101.7	−0.087
D	*Bna*C5DABB1	*BnaC05T0290400ZS*	22,643.11	5.12	34.97	108.3	0.046
E	*Bna*A3DABB1	*BnaA03T0156300ZS*	31,777.88	5.51	31.24	77.61	−0.286
E	*Bna*C3DABB1	*BnaC03T0182000ZS*	31,751.78	5.4	35.77	79.36	−0.286

## Data Availability

Data are contained within the article or Appendix A.

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
