# Peer review of "Genome-Wide Identification and Multi-Stress Response Analysis of the DABB-Type Protein-Encoding Genes in Brassica napus"

_ijms, 2024, doi:10.3390/ijms25115721_

Round 1
Reviewer 1 Report
Comments and Suggestions for Authors
This study elucidates the role of stress-responsive A/B Barrel Domain (DABB) proteins in Brassica napus, shedding light on their structural characteristics, regulatory elements, and functional significance. Understanding the involvement of BnaDABBs in organ development and stress responses expands our knowledge of plant biology and could inform strategies for crop improvement and disease resistance in Brassica crops. The way the study is designed and the analysis is done adds something important to the plant genomics and biotech fields. But, hitting on the points mentioned would make your paper even sharper, more connected, and convincing.
1) The VIGS construction should also be evaluated and discussed, because the members of other gene family might also be silenced.
2) In the section that talks about quantitative real-time PCR (qRT-PCR) analysis (pages 10-11), it'd help if the authors could throw in more details about the stats tests used to sort through expression data. Like, did ya use two-tailed tests and what was the significance level you were lookin' at (e.g., p<0.05)? This could really make your results clearer and more trustworthy.
3)Also, explaining why you picked the sample size for the qRT-PCR experiments would make the study seem more solid. Sayin' how you decided on the number of samples, maybe with a power analysis, could help with any doubts about your findings being strong enough.
4) in the last paragrah if introduction, it would be better to refer in third person instead of using "we".
5) When mentioning the 'cis-acting elements', the cis should be written in italic. Please revise the related part through the manuscript.
6) the writing quality needs improvement due to awkward sentence structures, questionable word choices, along with minor grammar and punctuation errors, all of which affect the clarity and readability of the text.
here are some examples:
Sentence structure: Original: "However, B. napus production is often threatened by fungal diseases, especially the Sclerotinia stem rot caused by Sclerotinia sclerotiorum [3], posing significant challenges to B. napus yield and economic costs [4]." Improved: "However, fungal diseases, particularly Sclerotinia stem rot caused by Sclerotinia sclerotiorum [3], often threaten B. napus production, posing significant challenges to both yield and economic costs [4]."
-
Word choice: Original: "B. napus, as an main planted-type of oilseed crop..." Improved: "B. napus, as a primary cultivated oilseed crop..."
-
Flow: Original: "However, B. napus production is often threatened by fungal diseases, especially the Sclerotinia stem rot caused by Sclerotinia sclerotiorum [3], posing significant challenges to B. napus yield and economic costs [4]. Therefore, it is of crucial importance for enhancing B. napus resistance to S. sclerotiorum." Improved: "However, B. napus production is frequently threatened by fungal diseases, with Sclerotinia stem rot, caused by Sclerotinia sclerotiorum [3], being particularly detrimental, impacting both yield and economic costs [4]. Consequently, enhancing B. napus resistance to S. sclerotiorum is of paramount importance."
-
Grammar: Original: "B. napus, as an main planted-type of oilseed crop..." Improved: "B. napus, as a main cultivated oilseed crop..."
-
Punctuation: Original: "Furthermore, the enrichment of numerous cis-acting elements in hormone induction and light response were revealed in the promoters of BnaDABBs." Improved: "Furthermore, the enrichment of numerous cis-acting elements in hormone induction and light response was revealed in the promoters of BnaDABBs."
Comments on the Quality of English Language
This study elucidates the role of stress-responsive A/B Barrel Domain (DABB) proteins in Brassica napus, shedding light on their structural characteristics, regulatory elements, and functional significance. Understanding the involvement of BnaDABBs in organ development and stress responses expands our knowledge of plant biology and could inform strategies for crop improvement and disease resistance in Brassica crops. The way the study is designed and the analysis is done adds something important to the plant genomics and biotech fields. But, hitting on the points mentioned would make your paper even sharper, more connected, and convincing.
1) The VIGS construction should also be evaluated and discussed, because the members of other gene family might also be silenced.
2) In the section that talks about quantitative real-time PCR (qRT-PCR) analysis (pages 10-11), it'd help if the authors could throw in more details about the stats tests used to sort through expression data. Like, did ya use two-tailed tests and what was the significance level you were lookin' at (e.g., p<0.05)? This could really make your results clearer and more trustworthy.
3)Also, explaining why you picked the sample size for the qRT-PCR experiments would make the study seem more solid. Sayin' how you decided on the number of samples, maybe with a power analysis, could help with any doubts about your findings being strong enough.
4) in the last paragrah if introduction, it would be better to refer in third person instead of using "we".
5) When mentioning the 'cis-acting elements', the cis should be written in italic. Please revise the related part through the manuscript.
6) the writing quality needs improvement due to awkward sentence structures, questionable word choices, along with minor grammar and punctuation errors, all of which affect the clarity and readability of the text.
here are some examples:
Sentence structure: Original: "However, B. napus production is often threatened by fungal diseases, especially the Sclerotinia stem rot caused by Sclerotinia sclerotiorum [3], posing significant challenges to B. napus yield and economic costs [4]." Improved: "However, fungal diseases, particularly Sclerotinia stem rot caused by Sclerotinia sclerotiorum [3], often threaten B. napus production, posing significant challenges to both yield and economic costs [4]."
-
Word choice: Original: "B. napus, as an main planted-type of oilseed crop..." Improved: "B. napus, as a primary cultivated oilseed crop..."
-
Flow: Original: "However, B. napus production is often threatened by fungal diseases, especially the Sclerotinia stem rot caused by Sclerotinia sclerotiorum [3], posing significant challenges to B. napus yield and economic costs [4]. Therefore, it is of crucial importance for enhancing B. napus resistance to S. sclerotiorum." Improved: "However, B. napus production is frequently threatened by fungal diseases, with Sclerotinia stem rot, caused by Sclerotinia sclerotiorum [3], being particularly detrimental, impacting both yield and economic costs [4]. Consequently, enhancing B. napus resistance to S. sclerotiorum is of paramount importance."
-
Grammar: Original: "B. napus, as an main planted-type of oilseed crop..." Improved: "B. napus, as a main cultivated oilseed crop..."
-
Punctuation: Original: "Furthermore, the enrichment of numerous cis-acting elements in hormone induction and light response were revealed in the promoters of BnaDABBs." Improved: "Furthermore, the enrichment of numerous cis-acting elements in hormone induction and light response was revealed in the promoters of BnaDABBs."
Reviewer 2 Report
Comments and Suggestions for Authors
The manuscript “Genome-wide identification and multi-stress response analysis of the DABB-type protein-encoding genes in Brassica napus” aims to characterize the biological features of rapeseed DABB gene family in response to abiotic and biotic stresses through a variety of analyses. The authors identified the whole family members using bioinformatic analysis and performed in silico analysis to examine their expression patterns. Several candidate genes, such as BnaC1DABB1, BnaA1DABB1, and BnaA1DABB2, were highlighted by their potentials in fungi resistance. Overall, the design and analysis of this study are sound, and the results are exhaustive. However, some results are required to re-examine or re-analyze. Below, I outlined some of my major concerns that need to be addressed before next submission.
ABSTRACT:
The full name of DABB, i.e. dimeric alpha+beta barrel, should be clearly provided at the beginning of the Abstract.
INTRODUCTION:
The authors try to emphasize the importance of BnaDABB genes in response to biotic stress. Thus, I suggest that they should inform the readers about the severity and damages of pathogen and fungal diseases in agricultural production, such as annual economical losses.
Has the DABB family been characterized in other plant species? If yes, a brief description should be provided.
RESULTS:
The authors used 4 out of 5 Arabidopsis DABB genes as queries to identify their orthologs in rapeseed. They should explain the exceptional AtDABB gene.
Figure 1, since only one panel involved, no need to list “A” at the up-left corner. The gene name should be provided together with the gene ID.
Figure 2A and C, the colored bars, gray background and colored lines should be described in the figure legend. B. napus, A. thaliana, B. carinata, B. juncea and C. sativa should be italic.
Figure 2B, what are the values indicated? Similarity or identity?
Line 131-133, significant differences were identified through sequence alignment and structural analysis. However, the authors should explicitly explain these differences.
Line 140, how to justify a “GOOD” or “BAD” hydrophilicity of protein?
Figure 3, one or two DABB domains were identified and used to categorize the family members. I suggest that the DABB domain should be labeled in 3D and E to illustrate the structural differences between groups.
Figure 3C, the authors outlined 11 conserved motifs in BnaDABB proteins. I suggest that they should present the sequences and the locations of these motifs to help readers understand the structural conservation of this family.
Figure 4 should be converted to a table to make it more readable.
Line 159-161, I don’t understand that how the author reveals the findings by STATISTICAL ANALYSIS.
The expression analyses of BnaDABB gene family are heavily relied on public repository data, which significant decrease the novelty of current study. Since the expression data have been provided in supplemental tables, figure 5-7 could be removed. Instead, the author could employ RT-qPCR method to investigate the dynamic transcriptional changes of the DABB genes not only in rapeseed, but also in B. rapa and B. oleracea.
Figure 8C, the same plant/leaf should be imaged to illustrate the development of lesion that caused by S. sclerotiorum infection.
Line 259, what are the differences that identified from the promoter regions of these three BnaDABB genes in two contrast genotypes.
M&M:
Line 342, how did the seeds germinate?
Line 400, the 2kb sequence of promoter, not CDS.
Line 403-404, if the expression data have been published, the reference(s) should be provided. Otherwise, the detailed procedure of the treatment and data analysis should be amended.
Line 417, the primer sequence and gene ID of BnaActin7 are missing.
Comments on the Quality of English LanguageModerate editing of English language required
Round 2
Reviewer 1 Report
Comments and Suggestions for Authors
The manuscript has been sufficiently improved, and I have no more suggestion.
Author Response
We genuinely thank the expert reviewers for your constructive and helpful suggestions and comments, which help us to improve the quality of our manuscript. Our responses in green font can be found beneath each original reviewer’s comment. A version of the revised manuscript with track changes is included for review.
We believe this revised version is much improved, and hope the manuscript is now suitable for publication in IJMS.
Reviewer 2 Report
Comments and Suggestions for Authors
The authors have clearly addressed the most of my concerns, and the quality of the manuscript has been improved significantly. However, I have a few additional suggestions to improve the readability of the manuscript.
1. Table 1, Title “physio-biochemical characterization of BnaDABB family members”; the members should be sorted by their chromosomal location not by molecular weight; geneID should be listed beside gene name, as a new column.
2. Figure 3, the legend of 3d is incorrect.
3. Figure 5a, the gene IDs are different from the ones listed in Table 1. Revision is required to make them in consistent.
4. Figure 6, promoter sequence comparison of the XXX gene between species A and B.
5. Line 376, provide the link of the B. napus (ZS11.v0) protein database
6. Gene ID of BnaACTIN7 gene is still missing.
Comments on the Quality of English LanguageMinor editing of English language required
